# Antibacterial Activity of Medicinal Plants and Their Constituents in the Context of Skin and Wound Infections, Considering European Legislation and Folk Medicine—A Review

**DOI:** 10.3390/ijms221910746

**Published:** 2021-10-04

**Authors:** Silvia Bittner Fialová, Katarína Rendeková, Pavel Mučaji, Milan Nagy, Lívia Slobodníková

**Affiliations:** 1Department of Pharmacognosy and Botany, Faculty of Pharmacy, Comenius University in Bratislava, Odbojárov 10, 832 32 Bratislava, Slovakia; rendekova18@uniba.sk (K.R.); mucaji@fpharm.uniba.sk (P.M.); nagy@fpharm.uniba.sk (M.N.); 2Institute of Microbiology, Faculty of Medicine and the University Hospital in Bratislava, Comenius University in Bratislava, Sasinkova 4, 811 08 Bratislava, Slovakia; livia.slobodnikova@fmed.uniba.sk

**Keywords:** plant secondary metabolites, skin infections, antimicrobial activity, EMA, HMPC, folk medicine, wound healing, skin inflammation, herbal drugs

## Abstract

Bacterial infections of skin and wounds may seriously decrease the quality of life and even cause death in some patients. One of the largest concerns in their treatment is the growing antimicrobial resistance of bacterial infectious agents and the spread of resistant strains not only in the hospitals but also in the community. This trend encourages researchers to seek for new effective and safe therapeutical agents. The pharmaceutical industry, focusing mainly on libraries of synthetic compounds as a drug discovery source, is often failing in the battle with bacteria. In contrast, many of the natural compounds, and/or the whole and complex plants extracts, are effective in this field, inactivating the resistant bacterial strains or decreasing their virulence. Natural products act comprehensively; many of them have not only antibacterial, but also anti-inflammatory effects and may support tissue regeneration and wound healing. The European legislative is in the field of natural products medicinal use formed by European Medicines Agency (EMA), based on the scientific work of its Committee on Herbal Medicinal Products (HMPC). HMPC establishes EU monographs covering the therapeutic uses and safe conditions for herbal substances and preparations, mostly based on folk medicine, but including data from scientific research. In this review, the medicinal plants and their active constituents recommended by EMA for skin disorders are discussed in terms of their antibacterial effect. The source of information about these plant products in the review is represented by research articles listed in scientific databases (Science Direct, PubMed, Scopus, Web of Science, etc.) published in recent years.

## 1. Introduction

It will soon be a hundred years since Alexander Fleming returned to his London laboratory and discovered penicillin. Since then, antibiotics have shown incalculable mental and material value in saving lives. However, along with the antibiotic era, a new threat called antimicrobial resistance emerged, which currently limits the successful completion of the centenary of the antibiotic era [1,2]. The current role of scientists around the world is to meet the challenge of discovering new sources of effective antimicrobial drugs or to design and synthesize them. Medicinal plants have been the most valuable source of molecules with therapeutic potential throughout the history of mankind. Folk medicine of each civilization is based on natural products and, nowadays, medicinal plants still represent an important pool for the identification of novel drug leads [3]. The trend of natural products for therapeutic use may be especially beneficial in the treatment of skin and wound infections, due to a good accessibility of these infected lesions for topical drug application.

Skin with its protective barrier role and sensory, secretory, and thermoregulatory functions, is the largest organ of the human body [4]. The keratinized skin surface and the skin glands ducts are regularly colonized by commensal bacteria with low virulence, such as coagulase-negative staphylococci, non-pathogenic corynebacteria and cutibacteria. The microbiota of skin below the waist may typically contain also Gram-negative enteric rods and enterococci. Opportunistic pathogenic microbes (such as *Candida* spp., *Malassezia* spp., or *Staphylococcus aureus*) and primarily pathogenic bacteria with high pathogenic potential (e.g., *Streptococcus pyogenes*) may also be present in the skin microbiota of healthy carriers [5,6]. The human skin, especially of people who are hospitalized and underwent antibiotic treatment, is not rarely colonized by important nosocomial agents, such as Gram-negative non-fermenting bacteria (*Pseudomonas aeruginosa*, *Acinetobacter baumannii*) or yeasts, including the newly emerged opportunistic pathogen *Candida auris* [7,8,9]. Colonizing microorganisms are frequently introduced to skin and the underlying tissue during smaller or larger incidents leading to breaks in the skin surface integrity. Skin abrasions and wounds can be contaminated also by microbes from external sources, such as water (e.g., mycobacteria, *Vibrio*, *Aeromonas*, *Pseudomonas*), soil (e.g., *Nocardia*, or *Clostridium*), or contaminated surfaces (e.g., *Streptococcus pyogenes*, or *Staphylococcus aureus*). Microbes from the oral cavity of animals (e.g., *Pasteurella*, *Spirillum*, *Streptobacillus*, *Capnocytophaga canimorsus*, *Bartonella henselae)* or humans (anaerobic non-spore-forming bacteria) can gain access to the bite-wounds [5]. Even if the majority of skin injuries contaminated by microorganisms are self-limiting and healing ad integrum, without the need of any medical care in the healthy immunocompetent people, more severe infections may develop depending on the virulence and the number of introduced microbes, the severity of skin damage, and on the host factors, such as the state of immunity, age, underlying diseases, and the presence of foreign bodies in the wound. In some cases, skin infection may spread to subcutaneous tissue, leading to severe life-threatening sepsis, demanding prompt medical intervention and hospitalization. Underlying conditions, such as vascular diseases, pressure necrosis, or diabetes, lead to ineffective clearance of microbes and to chronic skin and wound infections (infected venous ulcer, decubitus, or diabetic foot). The final result is a wide variety of skin infectious diseases with diverse extension, severity, and progression rate (Table 1), which may be complicated by the spreading of infectious agents to adjacent soft tissue and/or bones, or to the bloodstream [10,11,12].

Various types of skin infections may have various specific etiological agents (Table 1); however, two bacterial species are the leaders—*Staphylococcus aureus* and *Streptococcus pyogenes.* Bacterial species belonging to the group of Gram-negative enteric rods or Gram-negative non-fermenting bacteria are isolated less frequently; they are more typical for hospitalized patients suffering from various nosocomial wound infections [5,13,17].

Even if the pathogenesis of *acne vulgaris* is multifactorial, and it cannot be considered as a typical skin infection, bacteria still have an impact on the clinical course of this skin condition. *Acne vulgaris* is a chronic disease of sebaceous glands with complex etiopathogenesis and genetic predisposition. Hormonal factors lead to increased sebum production, which together with defective keratinization in the pilosebaceous unit and activity of bacteria, organized in biofilm and colonizing the pilosebaceous duct, have a comedogenic effect. The increased volume of sebum and the plugged sebaceous duct leads to sebaceous gland disruption with the release of sebum to the surrounding tissue, resulting in a reactive inflammatory response. *Cutibacterium acnes*, colonizing sebaceous ducts, participates in the pathogenesis of acne vulgaris through metabolic products, and by the release of enzymes, cell wall structures, and other molecules with biologic activity. They stimulate cell proliferation, influence the keratinocyte differentiation, and have proinflammatory activity. *Cutibacterium* lipases degrade sebum triacylglycerol to unsaturated fatty acids with comedogenic activity. Proteases disrupt the follicular walls, which further stimulates the immune cells and humoral factors of immunity, participating in the inflammatory and cell-mediated hypersensitivity reactions [14,18,19]. Even if *Staphylococcus aureus* can be isolated from the acne lesions of some patients, its role in the pathogenesis of acne is still not clear [15,20]. Gram-negative rods (such as *Escherichia coli*, *Pseudomonas aeruginosa*, *Serratia marcescens*, *Klebsiella* spp., or *Proteus mirabilis*) are mostly only secondary invaders colonizing the pilosebaceous unit after antibiotic therapy, and exaggerating the intensity of damage [16].

The skin can be infected not only by bacteria, but also by viruses, such as human herpesvirus 1 and 2, varicella-zoster virus, papillomaviruses, or by yeasts, filamentous and dimorphic fungi, as well as by parasites [21,22,23].However, the main therapeutic concern is represented by the therapy of infections caused by resistant bacterial strains [2]. Therefore, this review is focusing predominantly on the drugs and plant products with activity against the most frequent bacterial causative agents of skin and wound infections.

Therapy choice of skin and wound infections depends on the severity of the disease and on the host factors. Cleaning and disinfection, sometimes augmented by a small surgical intervention and/or a short-course local antibiotic treatment, are sufficient for small localized infections in immunocompetent individuals. In more severe infections, and in immunocompromised patients, a complex therapy is needed, which is based on the systemic application of antimicrobial drugs, thorough surgical debridement, and supportive complex therapy [5,12]. Foreign body removal, whenever possible, must also be performed [24]. In the past, there were no problems with the susceptibility of microorganisms to antibiotics and to the broad-spectrum antiseptic agents. Unfortunately, nowadays, the emergence and increased spreading of antimicrobial resistance represents a therapeutic challenge also in the field of skin and wound infections [25,26,27]. The situation starts to be unsafe also with mupirocin—the broadly used topical antimicrobial drug for treatment of staphylococcal skin infections and for decolonization of risk patients before an operation [28]. Even a higher danger is represented by polyresistant nosocomial strains infecting operation site wounds, venous catheter insertion sites, burn wounds, and chronic wounds, such as decubiti [5,29]. Additionally, although there is fortunately no problem with susceptibility of *Streptococcus pyogenes* to penicillin (the drug of choice), patients allergic to this drug may be threatened by increasing resistance of *S. pyogenes* to macrolides and lincosamides [30,31,32]. Plant products may in these cases provide help in the form of alternative, supportive, or combination therapy with the “classical” antimicrobial drugs.

Except for a direct antimicrobial effect, biologically active plant molecules can trigger the host’s own protective mechanisms by modifying the immune response; they may also protect cells and tissue from oxidative stress, and may stimulate healing and tissue regeneration [33]. When focusing on the antimicrobial effect of plant products, the direct inactivation of the target microbe is not the only mode of their antibacterial activity. By modification of the bacterial cell metabolism, by genes expression regulation, or by interference with various molecular targets in the bacterial cell, the active compounds of plants can decrease the virulence of the bacterial invader, or make it more vulnerable to antibiotic treatment, e.g., by interference with antimicrobial resistance mechanisms [34]. Moreover, the antimicrobial treatment of chronic skin and wound infections is complicated by biofilm produced at the site of infection by microbial agents. Therefore, the antibiofilm activity of antimicrobial remedies is also welcome [35,36].

In recent decades, the antimicrobial effect of many plant-extracts, as well as isolated plant substances, has been tested with promising results [37,38]. The antibacterial properties and antibacterial activity mechanisms of plant products with potential in the therapy of bacterial skin and wound infections are discussed below.

## 2. Medicinal Plants and Their Constituents with Antibacterial Activity Intended in the Treatment of Skin Disorders According to European Legislation

The therapy of skin and wound infections can be supported by application of natural products originating from a wide variety of medicinal plants. These products include whole plant-extracts, plants’ exudates, or isolated active substances. Small herbal substances that come into play in the process of healing skin infections may be involved by various mechanisms with different outcomes, including antimicrobial, antioxidant and anti-inflammatory activity; moreover, they can also augment the regeneration of damaged skin. The multitarget action of herbal medicines can facilitate the treatment of skin infections by acting on the important processes involved in the pathophysiology of these diseases [33].

The regulation in the field of herbal drugs (term “herbal drug” is used in European Pharmacopoeia, and pharmacognostical textbooks. EMA/HMPC uses the equivalent term “herbal substance”) in the European Union (EU) is under the authority of the European Medicines Agency (EMA) with its the Committee on Herbal Medicinal Products (HMPC). It is responsible for compiling and assessing scientific data on herbal substances, preparations, and combinations, to support the harmonization of the European market. HMPC evaluates all herbal drugs, and for each drug establishes EU monographs covering the therapeutic uses and safe conditions of well-established (WEU) and/or traditional use (TU) for herbal substances and preparations [39]. Currently recommended herbal substances by EMA for skin are listed in Table 2 [40]. Most of them are intended for TU, with the meaning of each defined in the EU Directive 2001/83/EC. EMA requires for WEU the demonstration of sufficient safety and efficacy data, but for TU the herbal substances are accepted on the basis of sufficient safety data and only plausible efficacy. Most of the medicinal plants listed in Table 2 are known from European folk medicine, but there are some plants (e.g., *Commiphora molmol* or *Melaleuca alternifolia*) that are not native to Europe, but they have a very long tradition in medicinal use in the countries of EU. These herbal substances are intended and designed for use without the supervision of a medical practitioner for diagnostic purposes or for prescription or monitoring of treatment; they are exclusively for administration in accordance with a specified strength and posology; they are an oral, external, and/or inhalation preparation [41,42]. EU herbal monographs provide all the information necessary for the use of a medicinal product containing a specific herbal substance or preparation including what the herbal product is used for; who the herbal product is intended for; safety information, such as information regarding undesirable effects, interactions with other medicines or duration of use [42].

Natural products, possessing both antibacterial and anti-inflammatory activities, are very convenient in relation to bacterial infections. Conversely, these plants’ secondary metabolites, mostly present only weaker antibacterial activities in comparison with antibiotics; however, some of them can reverse the resistance to antimicrobial agents in the bacterial cell [44]. According to Sychrová et al. (2020), antibacterial agents are considered as natural products, with MIC (minimum inhibitory concentration) up to 32 μg/mL and as anti-inflammatory agents, with IC_50_ (the half maximal inhibitory concentration) up to 15 μM [45].

The antibacterial activity of natural products can be explained by different mechanisms (Figure 1), depending on the spectrum and the content of compounds present in herbal drug or in final extract.

### 2.1. Coumarins

With the phenolic structure (benzopyran-2-one, or chromen-2-one derivatives) are coumarins, likewise flavonoids, good drug adepts thanks to a variety of pharmacological activities, including antimicrobial, antioxidant, and anti-inflammatory effects [48]. De Souza et al. (2005) tested 45 different coumarin derivatives on various bacterial species including *Staphylococcus aureus*. Of many tested coumarins, osthenol (Figure 2), which could be found in the plants of the genus *Angelica*, exhibited antibacterial activity and inhibited the growth of *Bacillus cereus* and *S. aureus* in MICs ranging from 62.5 to 125 µg/mL. It is suggested that the prenylation in C8 and OH group in C7 may increase the antibacterial activity of coumarins [49]. The activity of coumarins and coumarin derivatives on *S. aureus*, including methicillin-resistant *Staphylococcus aureus* strains (MRSA), is explained by the binding of coumarin moiety (Figure 2) to the B subunit of DNA gyrase [50,51]. Coumarins are major constituents in the plants of the genus *Melilotus*. Antibacterial activity of *Melilotus albus* was investigated by Stefanović et al. (2015). The ethanol, acetone, and ethyl acetate extracts of this plant were active against *B. subtilis* and *S. aureus* with MIC in the range from 1.25 mg/mL to 5 mg/mL [52].

### 2.2. Cinnamic Acid Derivatives

In plants, derivatives of cinnamic acid can be found in two chemical forms: as amides (phenolamides, such as avenanthramides) and as depsides (e.g., rosmarinic acid, salvianolic acids, or lithospermic acid). Compounds from both of these chemical forms revealed an antibacterial effect against Gram-positive, as well as Gram-negative bacteria [53]. Avenanthramides are the most known group of cinnamic acids amides to be found in *Avena sativa* L. EMA has recommendations for two herbal parts of *Avena*: Avenae herba and Avenae fructus, but only the community herbal monograph on dried fruit describes the indication in the treatment of minor inflammations of skin and in healing of minor wounds. While fruits are applied mostly on the skin, the herb is used for relieving mild symptoms of mental stress and to aid sleep. Moreover, folk medicine recommends both the oat fruits and herb for treatment of eczema or other skin defects. Oat fruits are typical also for the presence of proteins (glutelin, avenin), lipids (α- and β-avenotionin), steroid saponins, flavonoids (tricin, apigenin, luteolin-7-glucoside), and β-glucan [54]. Recently, several studies proved the anti-inflammatory effect and antibacterial activity of oat fruits on Gram-positive bacteria (including *S. aureus*) [55,56].

A large group of molecules form the group of amides based on putrescine, spermidine, or spermine (e.g., *N*^1^*,N*^10^-bis(dihydrocaffeoyl)spermidine, *N*^1^*,N*^14^-bis(dihydrocaffeoyl)spermine, *N*^1^*,N*^5^*,N*^14^-tris(dihydrocaffeoyl)spermine, or *N*^1^*,N*^5^*,N*^10^*,N*^14^*-tetra*-*p*-coumaroylspermine [43]. Information in available literature about antimicrobial properties of phenolamides is relatively rare and, in addition, the methodology differs among studies. As reported by Yingyongnarongkul (2008) bis-, tris- and tetra(dihydrocaffeoyl)polyamine conjugates showed antibacterial activity against vancomycin-resistant *S. aureus*. The MICs of the most effective tetra(dihydrocaffeoyl)polyamine conjugate ranged from 12.5 to 50 μg/mL [54]. As Roumani et al. mention, the most active antibacterial compounds, exhibiting a MIC value below 99 μg/mL, are dihydrocaffeoyl moieties. Anyway, the role of phenolamides in extracts with antibacterial activity is still unclear. Phenolamides are shown to also have anti-inflammatory activity in different cell cultures models, in molecular docking and in a murine itch model [57]. Several avenanthramides exhibited an inhibitory effect on TNF-α-induced NF-κB activation and NF-κB-mediated inflammatory response due to the downregulation of IKKβ activity in C2C12 skeletal muscle cell line [58,59]. Tri-*p*-coumaroylspermidine has been recently identified in the resin of *Salvia officinalis* flowers. This resin showed inhibitory activity against *Streptococcus mutans* and *Staphylococcus aureus* [60].

An important group of antibacterial molecules is represented by esters of caffeic acid with one of the most famous molecules, so-called “Lamiaceae tannin”, rosmarinic acid (Figure 3). Rosmarinic acid is a secondary metabolite widely distributed in plants, especially of the Lamiaceae family, where it presents one of the major phenolic compounds in polar extracts (water, alcohol-water). Antistaphylococcal activity of rosmarinic acid (MICs in the range from 2 to 15 μg/mL) was described in the past, together with its suppressive activity on the early stages of biofilm development, as well as a synergic effect with various antibiotics against resistant staphylococcal strains [61,62,63]. It has been found that not only rosmarinic acid, but also the other caffeic acid derivatives are expressing antibacterial effects, and their activities and mechanism of action have recently been reviewed in detail [64]. Mu et al. (2020) demonstrated that salvianolic acid A (Figure 3), a compound, which could also be found in many plants of Lamiaceae, inhibited sortase A activity, repressed adhesion of bacteria to fibrinogen, blocked the anchoring of protein A to the bacterial cell wall, inhibited the biofilm formation and the ability of *S. aureus* to invade adenocarcinoma human alveolar basal epithelial cells (A549) in vitro. Salvianolic acid A was revealed as a potential antivirulence agent to combat MRSA infections [65].

### 2.3. Flavonoids

Undoubtedly, flavonoids represent the largest group of natural molecules with antibacterial effects. They are daily consumed in fruits, vegetables, or nuts, seeds, stems, flowers, tea, wine, and honey. Nowadays, they have become more popular, not only in the scientific areas but especially in the general public, seeking “natural ways of healing”, which are favored [66]. Flavonoids are secondary metabolites, which, with very few exceptions, are present in all green plants and play a key role for the plant in protecting against pathogens. From a chemical point of view, it is a single group, with structures derived from benzo-γ-pyrone. More than 4000 structures are included in the basic subgroups [67], and it is even reported that there are more than 8000 molecules of them [68]. Flavonoids have a significant antibacterial effect, either alone, or in synergistic combinations. They destructively affect bacteria and/or, decrease their virulence by quorum quenching activity and act synergistically with conventional antibiotics, and increase the antimicrobial susceptibility of some bacteria e.g., by efflux pump inhibition [69,70,71]. The group of antibacterial flavonoids with MICs below 10 μg/mL against most common bacteria includes such substances as rutin, galangin, rhamnoisorobin, 2-hydroxylupinifolinol, *3*´-*O*-methyldiplacol, 2,8-diprenyleriodictyol, hesperetin, naringenin, pinocembrin, dihydrokaempferol, bartericin A, isobavachalcone, panduratin A, phloretin or licochalcone A [69]. Many flavonoids demonstrated mutual synergy as well [66,70]. Flavonoids are known to be effective both against Gram-positive and Gram-negative bacteria. The flavonoids’ antibacterial activity lies mainly in their interaction with bacterial cell membrane, where they disrupt phospholipid bilayers, inhibit the respiratory chain and ATP synthesis. Yuan et al. (2021) characterize two basic mechanisms of flavonoid interaction with the phospholipid bilayer: (i) interaction of hydrophilic flavonoids with polar phospholipid heads and (ii) intervention of more lipophilic flavonoids inside the phospholipid bilayer. More lipophilic flavonoids have a higher membrane affinity, thus, a higher activity [44]. However, other mechanisms have been described: from inhibition of nucleic acid synthesis, through blocking the fatty acid synthesis, to peptidoglycan synthesis inhibition. Flavonoids inhibit bacterial virulence by quorum quenching, which also impairs their ability to form biofilms [69]. In terms of antibacterial activity, hydroxylated structures are the more effective, especially 5,7-dihydroxyl substitution for flavone and flavanone, 2′ or 4′ hydroxylation for chalcones, and the hydroxyl group at 3 positions on the C ring of flavone [72]. Conversely, the methylation of -OH groups may decrease their activity, but hydrophobic substituents such as prenyl groups, alkylamino chains, alkyl chains, and nitrogen or oxygen-containing heterocyclic moieties usually enhance the antibacterial activity of all (semisynthetic) flavonoids [44,70]. As reported by Yuan and his team, the MICs of most flavonoids can be roughly calculated from their physicochemical parameters ACD/LogP or LogD_7.4_, and the lipophilicity is a key factor of plant flavonoids activity against Gram-positive bacteria [44]. Osonga et al. report that the antibacterial activity of flavonoids can also be favorably affected by slight modifications in their structure, e.g., by a phosphorylation [72]. If we focus on local skin and skin structures infections, flavonoids have shown inhibitory activity, especially against Gram-positive bacteria, including *Staphylococcus aureus* or *S. epidermidis*, which colonize or infect various skin injuries. Furthermore, they may act in synergy with β-lactams and probably also inhibit the activity of certain β-lactamases produced by bacteria [66,69,70].

It is generally known that flavonoids are an effective part of many preparations intended for the treatment of local infections and wound healing. Flavonoids such linarin, luteolin, 6-hydroxyluteolin, vicenin, genistein, catechin, ononin, quercetin, prunetin, or kaempferol showed a wound-healing effect in different wound models [73]. Furthermore, flavonoids such as liquiritigenin, naringenin, diosmetin, baicalein, quercetin, puerarin, chrysin, chamaejasmine and sulfuretin were tested and identified as natural molecules that can also reverse the pathological changes in atopic dermatitis thanks to their anti-inflammatory action [74].

Flavonoid rich herbal substances recommended by EMA to relieve skin inflammation and minimize the excessive perspiration of hands and feet include walnut leaves obtained from *Juglans regia* L., (Juglandaceae). Walnut leaves are popular in folk medicine. They are recommended for relieve the sunburn, or for treatment of acne and warts. The effective impact of various therapeutic forms obtained from *Juglans regia* leaves is substantiated with the presence of active compounds such as flavonoids (quercetin derivatives), tannins, and hydroxycinnamic derivatives [43]. Antibacterial properties of walnut leaves were examined by Vieira et al. (2019), who also compared the phytochemical and biological quality of green and yellow leaves. The green ones contained more active phenolic secondary metabolites (flavonoids and phenolic acids) and were more anti-inflammatory active. Moreover, both green and yellow leaves extracts were demonstrated to be active against clinical isolates of *Enterococcus faecalis*, *Listeria monocytogenes*, and methicillin-resistant *Staphylococcus aureus* [75]. The significant wound healing effect of *J. regia* leaves was proved by Baba & Qureshi (2021) [76]. Water-ethanol (70%) extract of the flavonoid-rich plant *Equisetum arvense* L. had antibacterial activity against Gram-positive cocci (incl. *Staphylococcus aureus* ATCC 29213 and clinical isolates of *Staphylococcus aureus*) in concentrations above 25 mg/mL [77]. The dry 50% ethanol extract of *E. arvense* stems (herb) was active against *Staphylococcus aureus* with MIC and MBC of 11.14 and 22.28 mg/mL, respectively [78]. Gendron with his team examined the antibacterial activity of *Urtica dioica* L. (nettle) and *Rosa nutkana* C. Presl against *S. aureus*, *Micrococcus luteus*, and *Pseudomonas aeruginosa*. For the testing, they used methanol extracts prepared by Soxhlet extraction. Only extract of *R. nutkana* was found to be effective against all tested bacteria. The extract of *U. dioica* has been shown to inhibit only *S. aureus* and *M. luteus* [79]. Antistaphylococcal activity of nettle extract rich in flavonoids (especially quercetin) and hydroxycinnamic derivatives was examined also in the study by Zenão et al. (2017): MICs of 70% ethanol extract of *U. dioica* against MSSA and MRSA ranged from 0.063 mg/mL to 0.500 mg/mL [80]. Petals of *Rosa damascena* are rich in the content of flavonoids, especially glycosides of quercetin and kaempferol [81]. Thanks to them, we have a polar extract of rose with an antibacterial and anti-inflammatory effect, useful in minor skin inflammations of bacterial origin. As reported by Maruyama et al. 2017, rose water inhibits the growth of *C. albicans* at a concentration of approximately 2.2%, reduces the viability of MRSA within 1 h, and decreases the neutrophil activation [82]. Folk medicine also recommends for skin disorders another flavonoid-rich medicinal plant—*Viola tricolor* L. of the family Violaceae, due to its anti-inflammatory, and immunomodulatory action [83]. The herb *V. tricolor* also possesses antibacterial activity. The MICs of herb extracts against *S. aureus* were found to be in the range from 0.15 to 2.5 mg/mL. Water, ethanol, and methanol extracts of *Viola* herb, were also active against some other Gram-positive and Gram-negative bacteria and against *C. albicans* [84].

### 2.4. Tannins

Tannins are very common secondary metabolites, found in many medicinal plants, especially in walnuts, cashew nuts, hazelnuts, and fruits such as grapes, blackberries, strawberries, mangoes, and in well-known beverages such as tea, coffee, and wine. They can be divided into two large groups: hydrolyzable and condensed tannins [85]. Similar to flavonoids, tannins are a very important group of natural antimicrobial substances. They inhibit multiplication and eradicate bacteria by various mechanisms, using their astringent properties (they precipitate metals and proteins). Tannins are able to chelate iron, important for bacterial metabolism, inhibit bacterial cell wall synthesis, disrupt the cell membrane, and inhibit the fatty acids biosynthetic pathways. Tannins have been reported to act as inhibitors of quorum sensing and by this way they can inhibit the expression of virulence factors, and interfere with biofilm formation. They are even considered a potential alternative to conventional antibiotics [86].

Hydrolyzable tannins include gallotannins and ellagitannins. The antibacterial effect of gallotannins is given by their structure, which also indicates their astringent action. A higher degree of galloylation and higher hydrophobicity results in stronger protein binding and a higher affinity for iron. Moreover, due to their large molecule size, they are unable to penetrate the bacterial membrane, therefore they are expected to preferentially interact directly with the bacterial membrane and membrane proteins [87]. It has been found that the black tea containing tannic acid may reduce methicillin-resistant *S. aureus* nasal and throat colonization and prevent biofilm formation [88]. A typical tannins-containing herbal drug, characterized especially by gallotannins content, is the Quercus cortex—a bark of *Quercus robur* L. (oak bark). This plant has a long tradition in folk medicine for chiefly external use. Oak bark is officially recommended by EMA for the treatment of local skin inflammation [40]. The antibacterial activity of oak bark was already documented in the past [89]. Recently, its anti-QS effect was also confirmed [90]. Ellagitannins, which may be found in red berries such as raspberry, cloudberry, or strawberry, have antibacterial activity as well. By testing different berries rich in tannins (esp. ellagitannins), an activity against *Staphylococcus aureus* was found [91]. The MIC of punicagalin from *Punica granatum* against *S. aureus* was established as 61.5 µg/mL. Different ellagitannins including agrimoniin (Figure 4), which is present in the Rosaceae family in genera such as *Agrimonia, Fragaria* or *Potentilla*, showed activity against different bacteria including *S. aureus* and, simultaneously, they did not exert a negative antibacterial effect on probiotic *Lactobacillus plantarum* [92]. In terms of skin disorders and wound healing, *Agrimonia eupatoria* L., agrimony, (Rosaceae) has a leading position in European official and folk medicine. The drug is mostly used orally in the form of herbal tea intended for gargling or in the form of infusion or decoction for oromucosal and cutaneous use, or as a bath additive. The biological activities of the agrimony herb are closely related to tannins (their content is around 11%) consisting of proanthocyanidins (dimers B1, B2, B3, B6, or B7, and trimers C1 or C2) and ellagitannins. Agrimoniin is present in the highest content. Except for tannins, the flavonoids (especially glycosides of apigenin, luteolin, quercetin, and kaempferol), phenolic acids (chlorogenic acid, ellagic acid, gentistic acid, vanillic acid, salicylic acid, and ferulic acid), and triterpenoids (ursolic acid, euscapic/tormentic acid esters) also play a significant role [93,94,95]. Effects of extracts (acetone, diethyl ether, ethanol, and water) of *A. eupatoria* were investigated on various Gram-positive and Gram-negative bacteria and fungi. The highest inhibitory activity on tested microorganisms was acquired by acetone extract. In terms of anti-biofilm activity, *A. eupatoria* extracts were examined on *Pseudomonas aeruginosa* and *Proteus mirabilis* strains. The acetone extract again proved to exhibit the best activity, while the water extract had no anti-biofilm activity on examined strains [96]. Ethanolic extract of agrimony made into an ointment showed antibacterial activity against *S. aureus* and significantly shortened the wound healing process in comparison to the control [97]. By EMA recommended *Hamamelis virginiana* L., (Hamamelidaceae) is rich especially in tri-, tetra—and pentameric proanthocyanidins. Cheesman et al. (2021) tested the antibacterial activity of different *H. virginiana* extracts on different species of streptococci and staphylococci. They found the methanolic and aqueous extracts effective in inhibiting the growth of *S. epidermidis* and *S. aureus*, with MIC values between 200 and 500 μg/mL [98].

### 2.5. Essential Oils

Essential oil (EO) in Latin “Aetheroleum”, could be defined as a mixture of volatile compounds predominantly of a terpenoid character (monoterpenes and sesquiterpenes) or of a phenylpropanoid one. In plants, EO can be localized and stored in specialized structures such as glandular trichomes, secretory ducts, and oil cells [67]. With plenty of biological effects (antimicrobial, analgesic, sedative, anti-inflammatory, spasmolytic, local anaesthetic, etc.), they are favored in wide use (food, cosmetics, fragrances, perfumery, pharmaceutical, and agronomic industries). Owing to their antimicrobial properties, many EOs are used as food or cosmetic preservatives [99]. The antimicrobial activity of EOs was extensively reviewed in the past [100,101]. Different targets for EOs in the bacterial cell were identified and described. By virtue of their lipophilic properties, they may influence the lipidic membranes of bacterial cells, and after they enter to the cell, the cytoplasm may be affected. In general, the reviewed antibacterial mechanisms of EO constituents include the cell wall disruption, effects on lipidic membrane permeability, reduction of membrane potential, disruption of proton pumps, depletion of ATP, inhibition of enzymes and other proteins production and secretion, inhibition of the overall bacterial metabolic activity, modification of the fatty acid outline to alter the cytoplasmic membrane, and the coagulation of cellular components [101,102,103,104,105]. Furthermore, EOs influence the bacterial virulence factors, e.g., by inhibition of toxins production and release from bacterial cell [106,107], and in addition, they might solve the bacterial resistance when used in mutual combinations or in combination with antibiotics [108,109]. EO components may possess not only antibacterial but even wound healing effects. (-)-Menthol (Figure 5), the major monoterpene in *Mentha × piperita*, and *M. canadensis* essential oils, possesses a wide spectrum of pharmacological activities including anti-inflammatory, antibacterial and wound healing [110]. According to EMA, peppermint (*Mentha × piperita* L., Lamiaceae) is intended for skin disorders, manifested as skin irritations. Antipruritic/anti-itching agents, such as menthol or peppermint oil, play an important positive role in skin disorders therapy [111]. Silva et al. (2019) proved that topical formulations based on menthol and saturated fatty acids could be suitable for wound healing. They show antibacterial activity against *Staphylococcus epidermidis* and *S. aureus,* including methicillin-resistant strains, and do not elicit any relevant cytotoxicity [112]. The monoterpene (-)-Menthol is a known TRPM8 (transient receptor potential cation channel subfamily M, member 8) agonist [110], but may also influence other types of TRP (transient receptor potential) channels (e.g., TRPV3 — transient receptor potential cation channel, subfamily V, member 3; TRPA1 — transient receptor potential cation channel, subfamily A, member 1) in a dose dependent manner [113,114]. The menthol interaction with TRP channels does not only result in analgesia or desensitization, but also in tissue regeneration [115]. Many EOs rich in 1,8-cineole, terpinen-4-ol, carvone, carvacrol, and thymol are excellent antibacterial, antiviral, or antifungal agents [101,116,117] with wide applications especially in cosmetics [118].

Many medicinal plants with EO rich in monoterpenes recommended by EMA are from the Lamiaceae family (*Mentha*, *Salvia*, *Origanum*) [40]. Except for the antibacterial properties of *Mentha* (described above), the antibacterial effect of *Salvia* is also very well known; its activity against MRSA was found to be comparable with antibiotics [119].

The herbal preparation Melaleucae aetheroleum (tea tree oil, TTO) has a long tradition in the treatment of skin disorders. EMA also lists this important traditional herbal medicinal product and recommends its usage. The systematic review by Casarin et al. (2017) points to various effects of TTO on periodontopathogens, dental plaque, gingivitis, periodontitis, and inflammatory responses [120]. Essential oil of *Melaleuca alternifolia* is rich in terpinen-4-ol, γ-terpinene, 1,8-cineole, α-terpinene, *p*-cymene, α-terpineole, aromadendrene, α-pinene, terpinolene, limonene, sabinene, α-phellandrene. In folk medicine, TTO is used on defects in the oral cavity, or for treatment of toothache and externally to cope with psoriasis [43]. Antibacterial, antioxidant, anti-inflammatory, and wound healing properties of TTO were proved by many studies over the last two decades [121,122,123].

There is evidence that some monoterpenes of TTO can exert higher antibacterial activity than complex EOs and should be considered as single antimicrobial agents. As an example, terpinen-4-ol (Figure 5) was more effective against MRSA and coagulase-negative staphylococci than TTO, and may be recommended as a single agent in the topical treatment of MRSA infections [124]. Monoterpenes such as (−)-borneol, (±)-camphor, carvacrol, L-carveol, L-carvone, *β*-citronellol, eugenol, *trans*-Geraniol, terpineol, and thymol exhibited better activity against *S. aureus* than conventional antibacterial drug sulphanilamide. Their MICs were in the range from 3 to 30 μg/mL and terpineol and thymol even showed bactericidal activity against *S. aureus* with MBC 120 μg/mL [125].

Not only monoterpenes, but also sesquiterpenic substances (Figure 5) show antibacterial activity and together with anti-inflammatory, antioxidant, and wound healing effects are intended for the treatment of skin disorders. Flower of *Matricaria recutita* L., (syn. *M. chamomilla*, Asteraceae), German chamomile, must not be missing in the report of plants used in skin disorders treatment. It is very often investigated due to its wide spectrum of traditional use. In the form of herbal tea/infusion, it is recommended for oral use or for inhalation. It is also used if the form of infusion for oro-mucosal or cutaneous application or as a liquid dosage form for bath additives. The main role in the therapy by *M. chamomilla* is attributed to its essential oil rich in sesquiterpenic (-)-α-bisabolol, chamazulene, and bisabololoxides A, B, C. Further secondary metabolites contributing to chamomile’s therapeutic effect are flavonoids (apigenin, quercetin, and luteolin derivatives), coumarins, and *N*^1^*,N*^5^*,N*^10^*,N*^14^*-tetra-p*-coumaroylspermine [43,126]. The antibacterial properties of *M. recutita* were investigated and confirmed in vitro and in vivo [126,127,128]. Furthermore, chamomile extract increased in vivo tissue granulation, density, and stimulated activation of fibroblasts and keratinisation in wounds [128]. Sesquiterpenes are active constituents also in EMA recommended herbal substances Millefolii flos and Milefolii herba [40]. Antibacterial properties of *Achillea millefolium* (yarrow) EO are well documented. The EO of *A. millefolium* aerial parts of French origin was tested against both Gram-positive and Gram-negative bacteria; EO of French *A. millefolium* was the most effective against *S. aureus* and *B. cereus,* with MICs 120 and 100 µg/mL, respectively [129]. The acetone/water extracts from flowering aerial parts of *A. millefolium* were in vitro active against *S. aureus* [130].

Despite all the positive properties of EOs in dermatologic usage, their allergenic potential remains questionable. Therefore, it is necessary to use them only in recommended concentrations and with individual concerns to avoid potential allergenic risk [118].

### 2.6. Diterpenoids

Diterpenoids form a large group of secondary metabolites, whose molecules are formally composed of four isoprene units joined in a head-tail form. Pimaranes, kauranes, labdanes, isopimaranes, abietanes, totarane, among others, have demonstrated antibacterial in vitro activity against various bacteria including *S. aureus* strains resistant to antimicrobial drugs. Some of the tested diterpenes showed MICs ranging from 8 to 25 μg/mL and may be considered as new antibacterial agents [131]. Carnosic acid (Figure 6) and carnosol are abietane diterpenoids with known antibacterial activity and both can be found in the genus *Rosmarinus* or *Salvia* [61,132]. Besides the activity against Gram-positive and Gram-negative bacteria and yeasts, carnosic acid may inhibit *Staphylococcus aureus* efflux pump by dissipation of the membrane potential and acts synergically with some conventional antibiotics such as gentamicin, ciprofloxacin, tetracycline, tobramycin, and kanamycin. This may be beneficial in the therapy of drug-resistant *S. aureus* strains [133,134]. The leaves of *Salvia officinalis* L. (Lamiaceae) are, among other indications, recommended by EMA for the relieving of skin, mouth, or throat inflammations. The leaves contain not only essential oil but likewise other constituents such as diterpenoids (carnosic acid, carnosol, rosmanol) and other compounds, owing to which, sage has antibacterial, anti-inflammatory, or wound healing properties [43,135]. Terpenes (including diterpenes and triterpenes) are also responsible for the antibacterial activity of Myrrh [43]. It was proved that hexane extract (5%) and essential oil (5%) of Myrrh reduced the viability of *S. aureus* 2–4 times in two hours. Thanks to the antibacterial effect, Myrrh is often added to creams or mouthwashes [136].

### 2.7. Saponins

Saponins are a diverse group of secondary metabolites widely distributed in plants all over the world. They contribute significantly to the antimicrobial activity of plants, as they are part of the plant’s defence against phytopathogens and herbivores [137]. They represent an important group of natural substances also in terms of antibacterial activity [138]. Saponins (e.g., ginsenosides, glycyrrhizic acid, β-aescin, α-hederin, hederacoside C, and primulic acid 1) can enhance the susceptibility of important multi-resistant bacterial strains (such as vancomycin-resistant enterococci or MRSA) to antibiotics [139]. Cucurbitacin B, belonging to the triterpenes, revealed an antibacterial effect against *S. aureus*, with MIC ranging from 0.12 to 0.44 μg/mL and had synergistic activity with tetracycline and oxacillin [140]. It has been demonstrated that triterpenic saponins seem to play important role in the wound healing process of the skin [141]. *Arctium lappa* L. (Asteraceae), commonly known as burdock, is traditionally used worldwide in phytotherapy. Different parts of the plant (dried roots, leaves, fruits, seeds) are used in folk medicine [142]. EMA recommends use of only the roots for medicinal purposes. They are rich in various active compounds, including triterpenes (α-amyrin and β-amyrin) [43]. The antibacterial effect of burdock roots has been proven by several studies [143,144]. Miazga-Karska et al. (2020) studied the antibacterial, antibiofilm, and antioxidant potential of low molecular weight peptides from burdock root and confirmed their anti-acne properties. Investigated preparations were effective against Gram-positive bacteria, while the Gram-negative bacteria remained untouched. This research brought promising results pointing to the antibacterial activity of examined extracts focusing on prospective anti-acne therapy [145]. From herbal substances rich in triterpene content, which are recommended by EMA for treatment of skin disorders, the mastic (a resin of *Pistacia lentiscus* L) should be mentioned. It contains triterpenes of tetracyclic euphane- and dammarane skeleton type and of the pentacyclic oleanane and lupane skeleton. Furthermore, it contains also tricyclic and bicyclic triterpenoids, some polyphenols, and volatile compounds [40]. There are several studies investigating the antibacterial properties of mastic, wherein most of them are dominated by the effect on *Helicobacter pylori* [146]. Nevertheless, antibacterial activity against skin pathogens for *P. lentiscus* is also documented. In a study provided by Mezni (2015), the fixed oil of *P. lentiscus* showed a bactericidal effect on *S. aureus* [147].

### 2.8. Carotenoids

Carotenoids are natural tetraterpenes and together with their oxygen derivatives, xanthophylls, belong to the group of natural pigments. [67]. Carotenoids have diverse functions and are considered to be health-promoting products [148]. In terms of skin, they have antioxidant and anti-inflammatory activities and accelerate wound healing [149,150]. They are important in photoprotection, aimed at the prevention of premature aging and skin cancer [151]. Several types of carotenoids also show antibacterial activity [152,153]. Together with their ability to promote wound healing, are carotenoids excellent candidates for topical treatment of infections. Carotenoid rich *Calendula officinalis* L. (marigold) flowers are used in folk medicine for the management of numerous diseases and disorders including minor fever symptoms, tonsillitis, high blood pressure, gastrointestinal disorders, skin inflammations, and wounds healing. However, not only carotenoids, but also other important compounds, such as triterpenes, flavonoids, coumarins, polysaccharides, and volatile compounds, are responsible for these therapeutic effects [40]. Medicinal products with marigold, owing to their antioxidant, antiseptic and anti-inflammatory properties, are intended for healing of skin lesions, dermal and mucosal pyogenic infections, burns, and wounds, [154]. Antibacterial activity of ethanol and methanol extracts of marigold were tested in vitro against different Gram-positive and Gram-negative bacteria. Overall, the highest efficiency was observed against *S. aureus* and *B. subtilis* [155]. Moreover, most of the studies on marigold are focusing on its anti-inflammatory and wound healing properties. Nicolaus with coworkers (2017) found that the *n*-hexanic and the ethanolic marigold extracts modulated the inflammatory phase of wound healing by activating the transcription factor NF-κB and by increasing the amount of the chemokine IL-8 [141]. The excellent wound healing properties of *C. officinalis* flowers were proved and reviewed in publications by Menda et al. and by Givol et al. [156,157].

### 2.9. Alkaloids

Despite their structural diversity, alkaloids are mostly solid, colourless substances that are sparingly soluble in water. Their solubility in water is increased by the formation of salts with organic acids or other acid-reactive metabolites [67]. Trigonelline (Figure 7) is a protoalkaloid, a pyridinium derivative from *Trigonella foenum-graecum* L. (fenugreek), but can be found also in coffee beans (*Coffea* sp.). Alongside with trigonelline, the seeds of *T. foenum-graecum* (Fabaceae) also contain a variety of other active compounds, including polysaccharides (galactomannans 25–45%), proteins and stilbenes, which together with flavonoids, steroidal saponins, and fatty oil contribute to reducing of minor skin inflammations [43,158]. Trigonelline is also known for its estrogenic effect and can be listed as phytoestrogen [159]. It is known that estrogen and compounds with estrogenic activity (incl. phytoestrogens) may be involved in the wound healing process through their action on estrogen receptors [160,161]. Trigonelline acts anti-inflammatory and has been reported to also have high anticandidal and antibacterial activity against some of the tested bacterial species. Unfortunately, it was not found to be active against MRSA, probably because *S. aureus* did not accumulate trigonelline [162]. Otherwise, fenugreek seeds from Algeria were effective with different strengths against various Gram-positive and Gram-negative bacteria. This effect was attributed to the water extract composition with high proportion of polyphenols [163]. The aerial part of *Solanum dulcamara* L., rich in alkaloids, was found to have antibacterial activity against *Streptococcus pyogenes*, *Staphylococcus epidermidis* and *S. aureus* [164]. Moreover, alkaloids found in other plants were investigated for their antimicrobial effects, and revealed antimicrobial activity against skin pathogens (including MRSA strains) [100]. Berberine and jatrorrhizine from *Mahonia aquifolium* (Pursh) Nutt. also showed antibacterial effects against clinical strains of *Cutibacterium acnes* from severe forms of acne, with MIC values between 5 and 50 µg/mL [165]. 8-hydroxylated benzo[c]phenanthridine alkaloids from *Chelidonium majus* L. exhibited antibacterial activity against MRSA strains, with MICs from 0.49 to 15.63 μg/mL [166]. Cushnie et al. (2014) reviewed several antibacterial effects of alkaloids: (i) inhibition of nucleic acid synthesis through activity on dihydrofolate reductase or topoisomerases, (ii) inhibition of bacterial cell division protein FtsZ, (iii) inhibition of bacterial enzymes (such as the sortase A), which leads to disruption of bacterial homeostasis, (iv) disruption of the outer membrane and the integrity of the cytoplasmic bacterial membrane. Alkaloids have also been shown to inhibit efflux pumps and affect numerous virulence factors [167].

### 2.10. Other Classes of Plant Constituents

Countless other natural substances could be applied in the treatment of skin infections, not all of which can be mentioned in details in this review. Several of them are outlined briefly below.

A soluble beta-1,3/1,6-glucan proved to increase the re-epithelialization and wound contraction in the mice model [168].

Heteropolysacharide pectin, which can be found in the terrestrial plant cell wall, seems to be effective especially in wound healing, as it can act as a scaffold for cell migration and differentiation. In addition, it has been found to kill both Gram-positive and Gram-negative bacteria [169]. Pectin, in combination with honey, in the form of biomedical hydrogels. acts in vivo as an effective agent for promoting and accelerating wound healing [170].

An important group of secondary metabolites with antimicrobial activity are glucosinolates, especially their degradation metabolites isothiocyanates (ITCs). Glucosinolates, the typical secondary metabolites in Brassicaceae family, are found in such plants as *Armoracia rusticana* or *Tropaeolum majus*. They are very popular and often used in folk medicine [171]. Kaiser et al. (2017) tested the antibacterial and antibiofilm activity of ITCs against *P. aeruginosa*. From the tested ITCs (allylisothiocyanate — AITC, benzylisothiocyanate, phenylethyl-isothiocyanate — PEITC, and their mixture — ITCM), the lowest MICs were recorded in AITC (103 μg/mL) and ITCM (140 μg/mL). Furthermore, the mixture of ITCs in sub-inhibitory concentrations also reduced biofilm mass and inhibited the metabolic activity of bacteria in mature biofilms [171]. The combinations of selected ITCs with conventional antibiotics were also examined [172]. AITC and PEITC in combinations with streptomycin or carbapenem had synergistic inhibitory activity on the growth of both Gram-positive (*S. aureus*, *Listeria monocytogenes*) and Gram-negative bacteria (*E. coli*, *P. aeruginosa*) [171,173,174].

Antimicrobial and anti-inflammatory properties were documented also in the case of phloroglucinol derivative hyperforin and naphtodiantrone hypericin of St. John’s Wort *(Hypericum perforatum* L., Hypericaceae). Hypericin and hyperforin are considered to be the main components responsible for the antimicrobial activity of St. John’s Wort. Both components were active against *S. aureus* strains, with MICs between 12.5 and 50 μg/mL [175]. Hyperforin was also shown to stimulate the growth and differentiation of keratinocytes [176]. St. John’s Wort has a long tradition of use in the treatment of skin infections. Except for direct inhibitory activity on bacterial growth, *H. perforatum* has also anti-quorum sensing and anti-biofilm activity [177,178,179,180]. The wound healing properties and anti-inflammatory action of St. John’s Wort herb, except for hyperforin and hypericin, are related also to flavonoid content (specifically to derivatives of quercetin) [43,181,182]. In a clinical study, a cream with *Hypericum* extract standardized to 1.5% hyperforin was effective in the treatment of mild and moderate atopic dermatitis [183]. As reported by Nayak et al., the oil of *Hypercium perforatum* positively contributes to the wound-healing process [182]. 

In skin disorders, the ω-6-unsaturated fatty acids have an irreplaceable role, as they improve tissue regeneration by modulating the cell migration and proliferation, and by decreasing inflammation and pain [184]. In addition, fatty acids exert antibacterial activity, as reported by Zheng et al. The antimicrobial mechanism of unsaturated fatty acids, such as linoleic acid, palmitoleic acid, oleic acid, or linolenic acid, is based on the inhibition of bacterial enoyl-acyl carrier protein reductase (FabI), which is an essential component in bacterial fatty acid synthesis [185]. The long-chain polyunsaturated fatty acids were active against *C. acnes* and *S. aureus* with MICs in the range from 32 to 1024 mg/mL [186]. Antibacterial fatty acids are found in such herbal substances as Oenotherae oleum from *Oenothera biennis* L., which is found to be effective in the treatment of atopic dermatitis or eczema [187,188,189].

## 3. Synergy of Plant Metabolites in Mixtures

Secondary metabolites of plants may act concurrently by different mechanisms and may have different targets [190]. In many cases, the single substance is less effective than the native extract owing to this substance at the same concentration. This fact can be explained by the synergistic effect of the components in the extract [191]. Research concerning synergy mechanisms has been underway for a long time and new combinations of drugs or herbal substances are being developed to achieve the optimal effect and minimize the side effects of therapy [192]. The most effective method of combating antimicrobial resistance seems to be a combination of classical antibiotics with natural products, where their synergistic effect is exploited [193]. Finding such a combination is financially and timewise easier than developing new synthetic antibiotics. By examining the combinations, ineffective antibiotics can be reintroduced into practice in an effective combination with natural products to fight resistance problems [191,193]. There are already many studies proving the synergistic effect of antibiotics and natural molecules or plant extracts [63,69,171,193,194,195]. A good example is the combination of *M. × piperita* EO and gentamicin, which resulted in growth inhibition of *B. cereus*, *B. subtilis*, *S. aureus* incl. MRSA, *E. faecalis*, *E. coli*, and *P. aeruginosa*. This combination allowed a significant reduction of the amount of gentamicin needed to inhibit mentioned bacteria [196]. Not only peppermint oil, but also other EOs were affiliated to high incidences of synergy [197]. Moreover, various single monoterpenes applied in combinations can offer different interactions including synergy, but they may also act in an additive or antagonistic manner, which could be non-negligible information. Bassolé and Juliani (2012) reviewed the antimicrobial effects of different combinations of EOs and their components. Phenolic monoterpenes and phenylpropanoids in a combination with other EOs components were found to increase the bioactivities of these mixtures and a combination of phenolics (such as eugenol) with monoterpenes alcohols (such as thymol and carvacrol) produced synergistic effects on several microorganisms [198]. From the combinations of single monoterpenes (carvacrol, thymol, 1,8-cineole, and *p*-cymene), the highest synergy was observed in a combination thymol/1,8-cineole and thymol/*p*-cymene, but also thymol and carvacrol together work synergistically [199]. The mutual interaction between two chemotypes of *Thymus vulgaris* L., one of carvacrol/thymol chemotype and one of linalool chemotype, has also been studied. The mixture of these two EOs exhibited additive antibacterial effects, which especially increased their kill rate [200]. When focusing on polyphenols, the synergistic effect was recorded in the combination of epigallocatechin gallate and quercetin. Both substances were found to have antibacterial activity, but in combination their activity against methicillin-susceptible and methicillin-resistant *Staphylococcus aureus* significantly increased. Therefore, this combination might be considered as a potential topical antimicrobial agent for the treatment of skin infections caused by drug-resistant *S. aureus* [201]. The synergy between ellagitannins and flavonols was reported by Tomás-Menor et al. (2015). They tested different combinations of the main constituents of *Cistus salviifolius* (myricetin/punicalagin, ellagic acid/quercetin-3-glucoside, and quercetin-3-glucoside/myricetin) against *S. aureus* [202]. Synergy is also well documented for the combination of *Camellia sinensis* extract with other plants extracts (e.g., *Juglans regia*) or antibiotics (e.g., oxacilline) against resistant bacterial strains, such as MRSA [203].

Overall, many natural products have antibacterial activity when used as a single agent, and may act synergistically in combination with another plant products, or can help to increase the antimicrobial effect of conventional antimicrobial drugs [196,199,204]. The rationale of combination therapy is also supported by the recommendations of the World Health Organization, according to which such therapy should be preferred over monotherapy for various infectious diseases, including tuberculosis, malaria, and AIDS. In patients suffering from the mentioned diseases, the combination therapy allows multiple target sites to be affected, which decreases the risk of resistant mutants emerging during the prolonged or life-long therapy [205].

## 4. Conclusions

Skin and wound infections are among the most common health problems affecting people of all ages. In addition to standard pharmacotherapy of skin and wound infections, medicinal plants and their constituents are often used as well. Nowadays, the effects of natural products from plants and the mechanisms of their activity have been studied experimentally, with results showing a complex effect of these products, useful in the therapy of skin and wound infections. Owing to various constituents, the final effects of herbal therapeutic products remain complex. Many plant extracts exert antibacterial and anti-inflammatory activity, along with wound healing properties. For skin disorders, EMA recommends traditional herbal medicinal products, which have already demonstrated sufficient safety data and plausible efficacy. The biologic activities of recommended herbal drugs and their constituents, with focus on their antibacterial effects, were reviewed in this article.

## Figures and Tables

**Figure 1 ijms-22-10746-f001:**
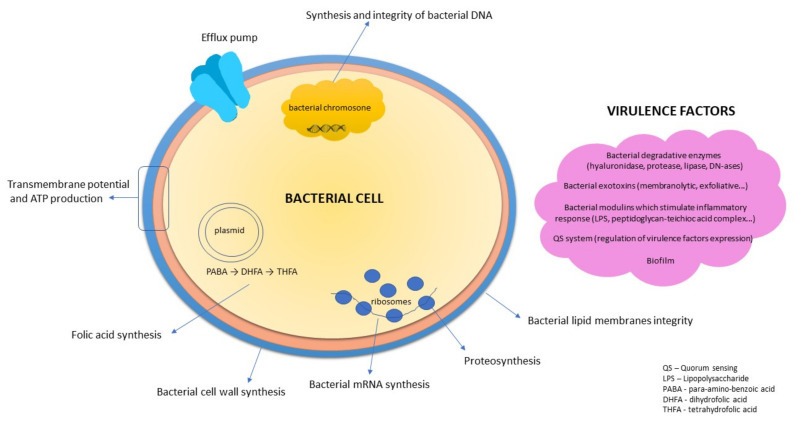
Targets for antibacterial compounds. (Figure according to Slobodníková, 2019) [46,47].

**Figure 2 ijms-22-10746-f002:**
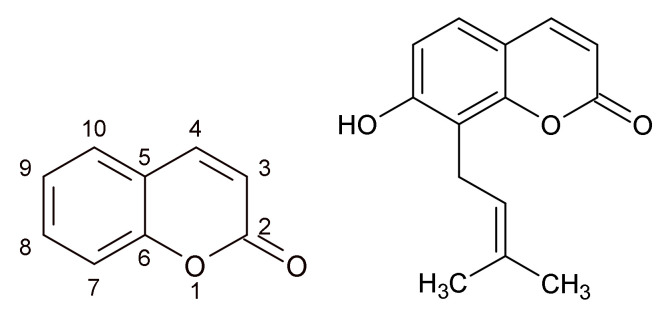
Coumarin (*2H*-chromen-2-one) (**left**) and osthenol (**right**).

**Figure 3 ijms-22-10746-f003:**
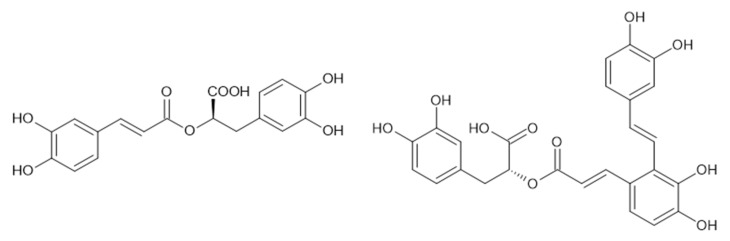
Rosmarinic acid (**left**) and salvianolic acid A (**right**).

**Figure 4 ijms-22-10746-f004:**
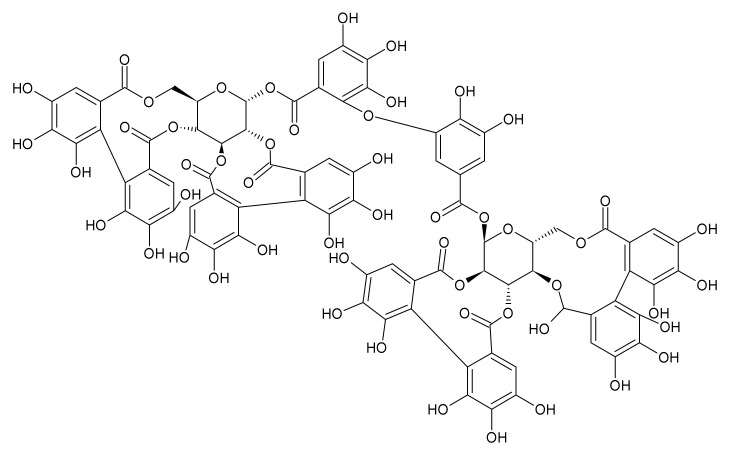
Agrimoniin.

**Figure 5 ijms-22-10746-f005:**
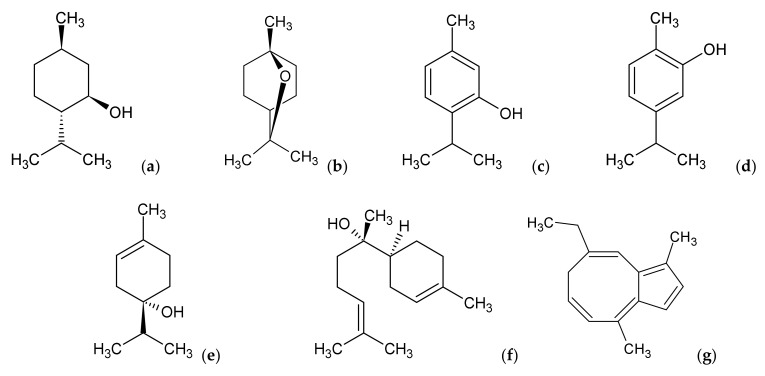
Selected monoterpenes: (−)-menthol (**a**), 1,8-cineole (**b**), thymol (**c**), carvacrol (**d**), terpinen-4-ol (**e**) and sesquiterpenes: (−)-α-bisabolol (**f**), chamazulene (**g**).

**Figure 6 ijms-22-10746-f006:**
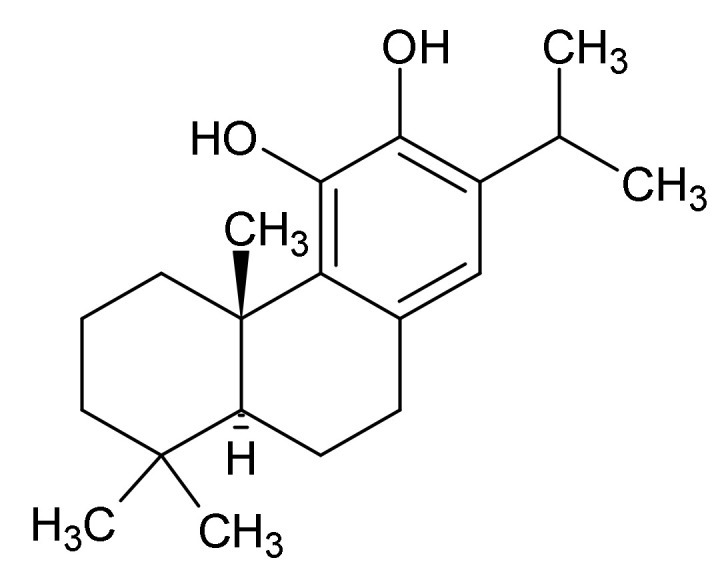
Carnosic acid.

**Figure 7 ijms-22-10746-f007:**
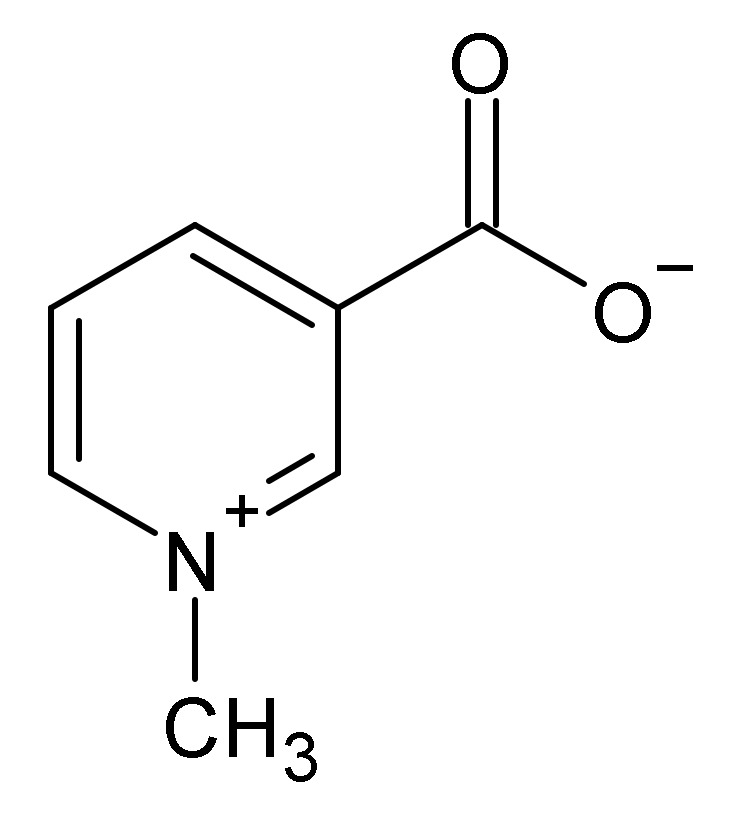
Trigonelline.

**Table 1 ijms-22-10746-t001:** The most frequent and most important skin and soft tissue infections (according to Esposito et al., 2016, Šimaljaková, 2019, Khorvash et al., 2012, Böni and Nehrhoff, 2003; modified) [13,14,15,16].

Type of Infection	The Most Common Infectious Agents
Impetigo	*Staphylococcus aureus*, *Streptococcus pyogenes*
Abscess	*S. aureus* Polymicrobial
Furuncle, carbuncle	*S. aureus*
Erysipelas	*S. pyogenes*
Cellulitis − common − injection drug use associated − fresh water injury associated − salt water injury associated − human bite associated − animal bite associated	*S. aureus*, *S. pyogenes* *S. aureus* *Aeromonas hydrophila* *Vibrio vulnificus* Human oral microbiota *Pasteurella multocida*; animal oral microbiota
Ecthyma gangrenosum	*Pseudomonas aeruginosa* Gram-negative bacilli
Surgical site infection—clean surgery	*Staphylococcus* spp.
Surgical site infection—all types	Anaerobic bacteria Enterococcus spp. Gram-negative bacilli *Staphylococcus* pp. *Streptococcus* spp.
Infected pressure ulcers	Polymicrobial
Necrotising fasciitis	*S. aureus, S. pyogenes* Polymicrobial
Myonecrosis	Myonecrotic clostridia
Acne lesions	*Cutibacterium acnes* (*Staphylococcus aureus*, Gram-negative rods)

**Table 2 ijms-22-10746-t002:** EMA recommended herbal medicinal products with therapeutic indication for skin (Traditional use).

Medicinal Plant (Source)	Drug (Herbal Substance or Preparation)	EMA Recommended Therapeutic Indication for Skin (Traditional Use) [40]	Secondary Metabolites Responsible for Intended Indications [40,43]
*Achillea millefolium* L.	herb/flower (Millefolii herba/Millefolii flos)	Traditional herbal medicinal product for the treatment of small superficial wounds.	sequiterpenic lactones, flavonoids, cinnamic acid derivatives
*Agrimonia eupatoria* L.	herb (Agrimoniae herba)	Traditional herbal medicinal product for relief of minor skin inflammation and small, superficial wounds.	tannins (agrimoniin), flavonoids, phenolic acids, triterpenoids
*Arctium lappa* L.	root (Arctii radix)	Traditional herbal medicinal product used in treatment of seborrhoeic skin conditions.	triterpenes (α-amyrin and β-amyrin), lignans (arctiin), and hydroxycinnamic derivatives
*Avena sativa* L.	fruit (Avenae fructus)	Traditional herbal medicinal product for the symptomatic treatment of minor inflammations of the skin (such as sunburn) and as an aid in healing of minor wounds.	avenanthramides (A–C), steroidal saponins (avenacoside A and B), triterpene saponins (avenacines), flavonoids
*Calendula officinalis* L.	flower (Calendulae flos)	Traditional herbal medicinal product for the symptomatic treatment of minor inflammations of the skin (such as sunburn) and as an aid in healing of minor wounds.	triterpenic saponins (calendulosides), triterpenic alcohols (α-amyrin, β-amyrin, lupeol), carotenoids (flavoxanthin, zeaxanthine, lutein), flavonoids (quercetin, isorhamnetin), coumarins, polysaccharides, volatile compounds
*Commiphora molmol* Engler and/or other *Commiphora* species	rubber resin (Myrrha, gummi-resina)	Traditional herbal medicinal product for treatment of minor wounds and small boils (such as furuncles).	diterpenes (muculol), triterpenes (myrrhanol A, myrrhanone A), steroids, essential oil
*Echinacea purpurea* (L.) Moench	fresh herb (Echinaceae purpureae herba, recens)	Traditional herbal medicinal product for treatment of small superficial wounds.	cinnamic acid derivatives, alkamides, polysaccharides
*Echinacea purpurea* (L.) Moench	root (Echinaceae purpureae radix)	Traditional herbal medicinal product used for the relief of spots and pimples due to mild acne.	cinnamic acid derivatives, alkamides, polysaccharides, glycoproteins, essential oil
*Equisetum arvense* L.	herb (Equiseti herba)	Traditional herbal medicinal product used for supportive treatment of superficial wounds.	flavonoids (glycosides of quercetin and kaempferol), silicates
*Glycine max* (L.) Merr.	oil (Soiae oleum raffinatum)	Traditional herbal medicinal product used for the symptomatic relief of dry skin conditions associated with mild recurrent eczema.	fatty acids (linoleic acid, oleic acid, palmitic acid, linolenic acid, stearic acid)
*Hamamelis virginiana* L.	leaf/bark (Hamamelidis folium/Hamamelidis cortex)	Traditional herbal medicinal product for relief of minor skin inflammation and dryness of the skin.	tannins, flavonoids, phenolic acids
*Hypericum perforatum* L.	herb (Hyperici herba)	Traditional herbal medicinal product for the symptomatic treatment of minor inflammations of the skin (such as sunburn) and as an aid in healing of minor wounds.	phloroglucinols (hyperforin with its derivatives: adhyperforin, furohyperforin, fu-roadhyperforin), naftodianthrone (hypericin), flavonoids (derivatives of quercetin), biflavons, tannins, volatile compounds
*Juglans regia* L.	leaf (Juglandis folium)	Traditional herbal medicinal product for the relief of minor inflammatory conditions of the skin and in excessive perspiration of hands and feet.	flavonoids (quercetin derivatives), tannins, hydroxycinnamic derivatives
*Matricaria recutita* L.	flower (Matricariae flos)	Traditional herbal medicinal product for adjuvant therapy of irritations of skin and mucosae in the anal and genital region and for the treatment of minor inflammation of the skin (sunburn), superficial wounds and small boils (furuncles).	essential oil, flavonoids (apigenin, quercetin and luteolin derivatives), coumarins, *N*^1^*,N*^5^*,N*^10^*,N*^14^-*tetra-p*-coumaroylspermine, polysaccharides
*Matricaria recutita* L.	essential oil (Matricariae aetheroleum)	Traditional herbal medicinal product used for adjuvant therapy of irritations of skin and mucosae in the anal and genital region, after serious conditions have been excluded by a medical doctor.	sesquiterpenes ((-)-α-bisabolol, chamazulene, and bisabololoxides A, B, C)
*Melaleuca alternifolia* (Maiden and Betch) Cheel, *M. linariifolia* Smith, *M. dissitiflora* F. Mueller or other species of *Melaleuca*	essential oil (Melaleucae aetheroleum)	Traditional herbal medicinal product for treatment of small superficial wounds and insect bites, for treatment of small boils (furuncles and mild acne) and for the relief of itching and irritation in cases of mild athlete’s foot.	terpinen-4-ol, γ-terpinene, 1,8-cineole, α-terpinene, p-cymene, α-terpineole, aromadendrene, α-pinene, terpinolene, limonene, sabinene, α-phellandrene
*Melilotus officinalis* (L.) Lam.	herb (Meliloti herba)	Traditional herbal medicinal product used for the treatment of minor inflammations of the skin.	coumarins (coumarin, scopoletin, umbelliferone, melilotin), flavonoids (glycosides of quercetin and kaempferol), triterpene saponins (melilotoside A-C)
*Mentha × piperita* L.	essential oil (Menthae piperitae aetheroleum)	Traditional herbal medicinal product used for the symptomatic relief of localized pruritic conditions in intact skin.	menthol, menthone, 1,8-cineol
*Oenothera biennis* L., *O. lamarckiana* L.	oil (Oenotherae oleum)	Traditional herbal medicinal product for the symptomatic relief of itching in acute and chronic dry skin conditions exclusively based upon long-standing use.	fatty acids (*cis*-linoleic acid, γ-linolenic acid, oleic acid, palmitic acid)
*Origanum dictamnus* L.	herb (Origani dictamni herba)	Traditional herbal medicinal product used for the relief of minor skin inflammations and bruises.	essential oil (thymol, carvacrol), flavonoids, triterpenes, cinnamic acid derivatives
*Origanum majorana* L.	herb (Origani majoranae herba)	Traditional herbal medicinal product used for relief of irritated skin around the nostrils.	essential oil (terpinen-4-ol, (+)-*cis-*sabinene hydrat, α-terpinene, γ-terpinene, p-cymene), flavonoids, triterpenes, cinnamic acid derivatives
*Pistacia lentiscus* L.	resin (resinum/mastic)	Traditional herbal medicinal product used for the symptomatic treatment of minor inflammations of the skin and as an aid in healing of minor wounds	triterpenes (mastic acid, isomastic acid, oleanolic acid, tirucall), monoterpenes, sesquiterpenes, polyphenols
*Quercus robur* L., *Q. petraea* (Matt.) Liebl., *Q. pubescens* Willd.	bark (Quercus cortex)	Traditional herbal medicinal product for the symptomatic treatment of minor inflammation of the oral mucosa or skin	tannins (gallotannins, ellagitannins, flavano-ellagitannins, procyanidinoellagitannin)
*Rosa gallica* L., *Rosa centifolia* L., *Rosa damascena* Mill.	flower (Rosae flos)	Traditional herbal medicinal product used for relief of minor skin inflammation.	flavonoids (glycosides of quercetin and kaempferol), anthocyanins (cyanidin-3-O-β-glucoside), proanthocyanidins, carotenoids, essential oil
*Salvia officinalis* L.	leaf (Salviae officinalis folium)	Traditional herbal medicinal product for relief of minor skin inflammations.	cinnamic acid derivatives (rosmarinic acid, salvianolic acids, lithospermic acid), diterpenoids (carnosic acid, carnosol, rosmanol) triterpenes (oleanolic and ursolic acid), flavonoids (derivatives of apigenin and luteolin), essential oil
*Solanum dulcamara* L.	woody nightshade stem (Solani dulcamarae stipites)	Traditional herbal medicinal product for the symptomatic relief of mild recurrent eczema.	steroid alkaloids, steroid triterpenes, tropane alkaloids
*Trigonella foenum-graecum* L.	semen (Trigonellae foenugraeci semen)	Traditional herbal medicinal product for the symptomatic treatment of minor inflammations of the skin.	alkaloids (trigonelline), steroidal saponins, flavonoids, oil
*Urtica dioica* L., *Urtica urens* L.	herb (Urticae herba)	Traditional herbal medicinal product used in seborrhoeic skin conditions	flavonoids, cinnamic acid derivatives, triterpenes, coumarins
*Viola tricolor* L., and/or subsp. *V. arvensis* Murray (Gaud), *V. vulgaris* Koch (Oborny)	herb with flowers (Violae tricoloris herba cum flore)	Traditional herbal medicinal product for symptomatic treatment of mild seborrhoeic skin conditions.	flavonoids (rutin, *C*-glycosides of luteolin and apigenin), mucilage, caffeic acid, carotenoids (9-*cis*-violaxanthin), cyclopeptides

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
