# Peer review of "Antibacterial Activity of Medicinal Plants and Their Constituents in the Context of Skin and Wound Infections, Considering European Legislation and Folk Medicine—A Review"

_ijms, 2021, doi:10.3390/ijms221910746_

Round 1

Reviewer 1 Report

Although the manuscript presents a good topic, there are some critical comments and concerns that need to be clarified and addressed before further consideration.

  1. The authors should prove that the reviewed plant species are used in European folk medicine, where most of the reviewed plant species are originated from tropical and subtropical regions. 
  2.  Certain types of virus infections can also cause skin diseases, such as herpesviruses. Therefore, I recommend the authors consider this point and utilize the newly published paper on natural products against herpes simplex viruses (type 1 and type 2) that cause skin infection (facial and genital areas) (doi: 10.3390/v12020154).
  3. The authors did not cover all plant-derived compounds that inhibit bacteria and viruses, which are known to induce skin infections. For example, Cucurbitacin B, a member of triterpenes, was found to inhibit Staphylococcus aureus clinical strains and exhibits antiviral activity against HSV-1. I recommend the authors add this information based on the published paper (DOI: 10.1016/j.sajb.2016.10.001)
  4. Figure 1 presents bacterial cell targets for natural compounds and the authors provided an example of only Gram-positive bacteria Staphylococcus aureus. The manuscript contains information on skin infections caused by various types of fungi and bacteria (Gram-positive and Gram-negative). Therefore, it would be better to create other figures concerning fungi and Gram-negative bacteria. Alternatively, such information can be merged in one figure. 
  5.  The authors stated in the abstract section that ''In addition, natural products contribute to wound healing process''. The authors should discuss this point in the manuscript. It is known that natural products have anti-inflammatory properties using various mechanisms of action and can affect some enzymes and proteins that contribute to the wound healing processes. Please, consider this point and discuss it. 
  6. It would be better to add a new section that provides information about the most effective plant-derived substances that could be used as templates for further development of drugs useful in the therapy of skin infection.
  7. Finally, I recommend the authors double-check the whole manuscript for typing and grammatical errors. 

Author Response

Dear reviewer, thank you for the time you paid to review our manuscript, as well as for all of your suggestions on how to improve its text.

The authors should prove that the reviewed plant species are used in European folk medicine, where most of the reviewed plant species are originated from tropical and subtropical regions. 

Thank you for your comment. We have specified this in the text, please see section 2. Medicinal plants and their constituents with antibacterial activity intended in the treatment of skin disorders according to European legislation.

Certain types of virus infections can also cause skin diseases, such as herpesviruses. Therefore, I recommend the authors consider this point and utilize the newly published paper on natural products against herpes simplex viruses (type 1 and type 2) that cause skin infection (facial and genital areas) (doi: 10.3390/v12020154).

Definitely, you are right. Thank you for this comment, but in this article, we decided to review only antibacterial action, as the main problem, which made us write the review is the emerging and increasing rate of antimicrobial resistance. Herpetic viruses are very common as the cause of skin disorders and many of mentioned natural products are effective against HSV 1 or 2. If we include all relevant data about antiviral effects of mentioned herbal substances we will create a new long article as long as we have till now. In addition, such an article should cover also the herbal substances active against the other viruses infecting the skin, let's mention at least a very important group of human papillomaviruses. So, it could be a great thesis for the separate work.

The authors did not cover all plant-derived compounds that inhibit bacteria and viruses, which are known to induce skin infections. For example, Cucurbitacin B, a member of triterpenes, was found to inhibit Staphylococcus aureus clinical strains and exhibits antiviral activity against HSV-1. I recommend the authors add this information based on the published paper (DOI: 10.1016/j.sajb.2016.10.001)

Thank you for the information. We included this information in the text.

Figure 1 presents bacterial cell targets for natural compounds and the authors provided an example of only Gram-positive bacteria Staphylococcus aureus. The manuscript contains information on skin infections caused by various types of fungi and bacteria (Gram-positive and Gram-negative). Therefore, it would be better to create other figures concerning fungi and Gram-negative bacteria. Alternatively, such information can be merged in one figure. 

Thank you for the comment. As we specified the article only to bacteria, we decided to change Figure 1 to cover both the Gram-positive and the Gram-negative bacterial targets for natural products.

The authors stated in the abstract section that ''In addition, natural products contribute to wound healing process''. The authors should discuss this point in the manuscript. It is known that natural products have anti-inflammatory properties using various mechanisms of action and can affect some enzymes and proteins that contribute to the wound healing processes. Please, consider this point and discuss it. 

As we already mentioned above in our response to your comment on the viral skin infectious agents and antiviral herbal substances, wound healing is another complex and extensive field of medicine. Information about medicinal plants supporting would healing could form a separate review article. Of course, we don´t want to skip wound healing, as it is a very important component of our review topic, but we deleted some sections and kept the wound healing only informative, as an added value of plants used in the skin infections healing. It was not clearly stated in the manuscript, so now we better specified its aim.

It would be better to add a new section that provides information about the most effective plant-derived substances that could be used as templates for further development of drugs useful in the therapy of skin infection.

We included the active substances in Table 2 and then we also applied all herbal substances into discussed groups of secondary metabolites according to “the most active” constituents. We also extended the data about the antibacterial action of each medicinal plant. 

Finally, I recommend the authors double-check the whole manuscript for typing and grammatical errors. 

Thank you, we checked the manuscript again.

Reviewer 2 Report

In regards to essential oils, I am seeing several other plants have to be included in this review. Anyway good writing 

Author Response

In regards to essential oils, I am seeing several other plants have to be included in this review. Anyway good writing 

Thank you for your positive feedback and recommendation. As this work deals with the herbal drugs for skin disorders recommended by EMA (HMPC), all plants are mentioned (Table 2), including aromatic plants (major constituent essential oil).

Reviewer 3 Report

Dear authors,

the review titled “Medicinal Plants in Skin and Skin Structures Infections and Disorders Considering European Legislation and Folk Medicine – a Review” have an interesting topic but I suggest some changes to improve the quality of the paper. The main weakness is that is not clear the topic of the paper: the antimicrobial properties of natural compound or the use of natural compound in the management of skin disorders?

The abstract need to be deeply revised. In particular title and abstract doesn’t present any concordance. If I read only the abstract without the title I understand that the topic of the paper was the antibiotic activity of natural compounds, in relation with traditional medicine.

the manuscript presents three main parts.

In the first part you describe the issues related the management of bacterial infection and the resistance problem. Then, you describe the issues related the infection in skin and wound healing process, and at the end the skin disorders. In the second part you describe the antibacterial activity of selected natural compound, divided for chemical category, and some information about their uses for skin disorders. Traditional uses of medicinal plant in Europe are describe in the last part.

I think that the topic discussed in this order are not clear for the readers. I suggest to clarify if you discuss antibiotic activity of natural compound or skin related action of natural compound. The topics are connected but data are reported mixing the information.

I suggest to remove the second section (2.1, 2.3, 2.4, 2.5 etc 2.9) and move these data, in appropriated way, in the section of the traditional medicinal plants. As example when you discuss the traditional uses of marigold you can add a paragraph regarding the carotenoid action in skin related disorders. I suggest to report literature data in detail, selecting more relevant literature. In some part there are too sharp passage between one topic and others (for example 2.2 paragraph).

Please add a section related plants products active in the treatment of atopic dermatitis and seborrheic dermatitis, there is some reference but not reported data.

I suggest to deeply revised some part with too general information. The introduction can be reduced, as example line 31-54. Please rephrase paragraph 87-92.

Line 111-116: please change this part end explain what are the rational to use plant products as “new therapeutic options”, and add reference.

Please avoid expression like “ideal targets” line 113, “very promising antibacterial effect” line 230 231 “potential alternative to conventional antibiotics” line 337. “.. strong antimicrobial as well.” Line 347 “showed good activity” line 351. Please be precise and add information to avoid misunderstanding.

Please be sure to add appropriate reference at the end of each sentence.

In the present form the manuscript presents many weaknesses and in my opinion is not suitable for the journal IJMS

Author Response

Dear reviewer, thank you for the time you paid to review our manuscript, as well as for all of your suggestions on how to improve the text.

Dear authors,

the review titled “Medicinal Plants in Skin and Skin Structures Infections and Disorders Considering European Legislation and Folk Medicine – a Review” have an interesting topic but I suggest some changes to improve the quality of the paper. The main weakness is that is not clear the topic of the paper: the antimicrobial properties of natural compound or the use of natural compound in the management of skin disorders?

The abstract need to be deeply revised. In particular title and abstract doesn’t present any concordance. If I read only the abstract without the title I understand that the topic of the paper was the antibiotic activity of natural compounds, in relation with traditional medicine.

the manuscript presents three main parts.

In the first part you describe the issues related the management of bacterial infection and the resistance problem. Then, you describe the issues related the infection in skin and wound healing process, and at the end the skin disorders. In the second part you describe the antibacterial activity of selected natural compound, divided for chemical category, and some information about their uses for skin disorders. Traditional uses of medicinal plant in Europe are describe in the last part.

I think that the topic discussed in this order are not clear for the readers. I suggest to clarify if you discuss antibiotic activity of natural compound or skin related action of natural compound. The topics are connected but data are reported mixing the information.

I suggest to remove the second section (2.1, 2.3, 2.4, 2.5 etc 2.9) and move these data, in appropriated way, in the section of the traditional medicinal plants. As example when you discuss the traditional uses of marigold you can add a paragraph regarding the carotenoid action in skin related disorders. I suggest to report literature data in detail, selecting more relevant literature. In some part there are too sharp passage between one topic and others (for example 2.2 paragraph).

As you recommended, we deeply revised the abstract and introduction with aims to be clearer.

Regarding the second section: Thank you very much, you are right regarding the consistency of the text. This is also one of the possibilities and at the beginning, we considered that we would focus only on medicinal plants/herbal substances, but we encountered problems. I will try to explain what led us to focus mainly on groups of secondary metabolites.

  1. Calendula officinalis is an excellent example of a plant that has undergone extensive research and is an important plant in folk medicine. EMA assessment for the drug Calendulae flos has 37 pages. Nevertheless, the relevant information of this plant/herbal drug about the antibacterial effect from last years is not so rich. Let´s look at Violae herba, EMA assessment has 19 pages in many ways relevant research is missing. Information about these plants would be very uneven if we consider all areas (somewhere we find the data, somewhere we don't). In the case of the antibacterial effect of Violae herba, there are few older works, anyway, the data are not very satisfying.
  2. Since the article is focused on an antibacterial activity (we improved the aim, to be clearer), it would be difficult to explain the mechanism of action at the molecular level if considering only plant extracts.
  3. Then there is another aspect that has convinced us that characterizing the groups of constituents in EMA herbal drugs is the best way. Many would repeat in the text (as for example quercetin, or 1,8-cineole) which are very often presented in different plants. It would be strange to give everything to one plant and nothing to another or to repeat it.

Following your recommendations, we have modified the entire text and merged parts 2 and 4 to make it more consistent. The technical side is supplemented and we have also added the relevant active substances to each plant in Table 2.

We absolutely understand that mentioned plants and their constituents should contain more information and details from different points of view. Anyway, we had to "shorten" the review to focus only on antibacterial activity, because it is not realistic to address all aspects in detail and to be exact in each area: anti-inflammatory activity, wound healing, antiviral activity, antifungal activity (or to pay detailed attention to seborrheic dermatitis, atopic dermatitis, psoriasis, eczema, acne…) of so many plants. The range of the review articles for IJMS is quite limited. 

Now the second part explains which constituents are responsible for the antibacterial action of herbal substances in skin disorders recommended by EMA. This is the very specific view that has not been reviewed from this point of view yet and it is unique and very valuable in terms of pharmacognosy, pharmacy, and medicine. Further relevant data to each herbal substance have been added.

Please add a section related plants products active in the treatment of atopic dermatitis and seborrheic dermatitis, there is some reference but not reported data.

As we have already explained above, we had to "shorten" the review to focus only on antibacterial activity, because it is not realistic to address all aspects of so many plants in the connection with so complex topics as are the multifactorial and autoimmune skin disorders, such as atopic dermatitis, psoriasis or rosacea, and seborrheic dermatitis. These topics may be candidates for an individual review. Accordingly, we made changes in the text of our current review, being under evaluation for publication.

I suggest to deeply revised some part with too general information. The introduction can be reduced, as example line 31-54. Please rephrase paragraph 87-92.

The introduction was reduced, the paragraphs mentioned were rephrased.

Line 111-116: please change this part end explain what are the rational to use plant products as “new therapeutic options”, and add reference.

As the rationale why to use plant products in the therapy of skin infections is explained at the end of the first part of the review and in the introduction to the 2nd part, the lines 111-116 were deleted.

Please avoid expression like “ideal targets” line 113, “very promising antibacterial effect” line 230 231 “potential alternative to conventional antibiotics” line 337. “.. strong antimicrobial as well.” Line 347 “showed good activity” line 351. Please be precise and add information to avoid misunderstanding.

Thank you for your recommendation. We deleted controversial phrases.

Please be sure to add appropriate reference at the end of each sentence.

OK. Each information from available literature is properly cited.

In the present form the manuscript presents many weaknesses and in my opinion is not suitable for the journal IJMS

We worked really hard on the new version and we believe, you will be satisfied with all improvements.

Round 2

Reviewer 1 Report

The manuscript has been significantly improved.

Author Response

Dear reviewer,

thank you very much for your time to review our revised manuscript and for your positive feedback.

Reviewer 3 Report

Dear author,

The manuscript was deeply revised, and the quality of the work improved.

I suggest to deeply revised language and style. I suggest some changes in the last part of the manuscript

Please rephrase the conclusion. Conclusion should be connected to abstract information, please revise it

I report some suggestion reposted below

378 “herbal substances” please change this part

390 considering that you refer to C7 and C8 position of coumarins please add position numbers in corresponding figure.

698 please rephrase the sentence.

701 use TTO

709 “act antibacterial” please rephrase the sentence.

746 “Of these diterpenoids with antibacterial activity, carnosic acid (Fig. 6) and carnosol are important, both abietanes to be found in the genus Rosmarinus officinalis or Salvia” please rephrase the sentence.

760 “Myrrh extracts are often applied in creams or mouthwashes for the bactericidal effect [136].” Please improve English form

769 780 please reduce this paragraph

812 820 please reduce this paragraph

897” In combination with honey was pectin used to produce biomedical hydrogel, which has been shown in vivo as an effective agent for 898 promoting and accelerating wound healing” please improve English form

904 please use abbreviation only when there are repetitions. “ From the tested ITCs: 904 allylisothiocyanate (AITC), benzylisothiocyanate (BITC), phenylethyl-isothiocyanate 905 (PEITC) and their mixture (ITCM), was the lowest MIC was recorded inat AITC and in 906 ITCM, 103 μg/mL and 140 μg/mL respectively”

911 “very promising results” please change and add detail on this regards

912 “conventional” please rephrase the sentence, the meaning is not clear

928 “Used as a part of the preparation Hyperici oleum they contribute to the wound-healing process: please improve English form

932 act antibacterialy, please change like “exert antibacterial activity”

945- 954 please add reference for each sentence

956 “There are already many studies proving the synergistic 956 effect of antibiotics and natural molecules or plants extracts” please add at least 3 references.

966 As they mention please delete

967 “ with other components” please be specific other natural or synthetic compunds?

970 Of the single monoterpenes carvacrol, thymol, 1,8- cineole, and p-cymene, the highest synergy was observed in a combination thymol:1,8- 971 cineole and thymol:p-cymene, but also thymol and carvacrol together act synergically. Please rephrase

Conclusions: please improve this part focusing on antimicrobial activity

Author Response

Dear reviewer,

We are very thankful for your valuable comments on our manuscript. We appreciate a lot the time and work you paid to it. We accepted all recommendations, rephrased unclear parts, and included additional information, as you can see below.

378 “herbal substances” please change this part

Ok, done.

390 considering that you refer to C7 and C8 position of coumarins please add position numbers in corresponding figure.

OK

698 please rephrase the sentence.

OK, done.

701 use TTO

OK

709 “act antibacterial” please rephrase the sentence.

Changed to “show antibacterial activity”

746 “Of these diterpenoids with antibacterial activity, carnosic acid (Fig. 6) and carnosol are important, both abietanes to be found in the genus Rosmarinus officinalis or Salvia” please rephrase the sentence.

Yes, it was changed to be more understandable.

760 “Myrrh extracts are often applied in creams or mouthwashes for the bactericidal effect [136].” Please improve English form

Done.

769 780 please reduce this paragraph

The paragraph has been reduced to contain only brief general information.

812 820 please reduce this paragraph

The paragraph has been reduced to contain only brief general information.

897” In combination with honey was pectin used to produce biomedical hydrogel, which has been shown in vivo as an effective agent for 898 promoting and accelerating wound healing” please improve English form

OK, done.

904 please use abbreviation only when there are repetitions. “ From the tested ITCs: 904 allylisothiocyanate (AITC), benzylisothiocyanate (BITC), phenylethyl-isothiocyanate 905 (PEITC) and their mixture (ITCM), was the lowest MIC was recorded inat AITC and in 906 ITCM, 103 μg/mL and 140 μg/mL respectively”

OK

911 “very promising results” please change and add detail on this regards

Ok, done.

912 “conventional” please rephrase the sentence, the meaning is not clear

it was rephrased

928 “Used as a part of the preparation Hyperici oleum they contribute to the wound-healing process: please improve English form

OK, done.

932 act antibacterialy, please change like “exert antibacterial activity”

Ok, done.

945- 954 please add reference for each sentence

References were added

956 “There are already many studies proving the synergistic 956 effect of antibiotics and natural molecules or plants extracts” please add at least 3 references.

OK, references were added

966 As they mention please delete

OK, done.

967 “ with other components” please be specific other natural or synthetic compunds?

Specified to other essential oils components

970 Of the single monoterpenes carvacrol, thymol, 1,8- cineole, and p-cymene, the highest synergy was observed in a combination thymol:1,8- 971 cineole and thymol:p-cymene, but also thymol and carvacrol together act synergically. Please rephrase

OK, done

Conclusions: please improve this part focusing on antimicrobial activity

OK, the conclusion was deeply revised.

Round 3

Reviewer 3 Report

Dear authors, 

thanks for changes in the last version manuscript.

Now the manuscript is suitable for IJMS